# Emotional State Classification from MUSIC-Based Features of Multichannel EEG Signals

**DOI:** 10.3390/bioengineering10010099

**Published:** 2023-01-11

**Authors:** Sakib Abrar Hossain, Md. Asadur Rahman, Amitabha Chakrabarty, Mohd Abdur Rashid, Anna Kuwana, Haruo Kobayashi

**Affiliations:** 1Department of Computer Science and Engineering, Brac University, Dhaka 1212, Bangladesh; 2NSU Genome Research Institute, North South University, Dhaka 1229, Bangladesh; 3Military Institute of Science and Technology (MIST), Department of Biomedical Engineering, Dhaka 1216, Bangladesh; 4Department of EEE, Noakhali Science and Technology University, Noakhali 3814, Bangladesh; 5Division of Electronics and Informatics, Gunma University, 1-5-1 Tenjin-cho, Kiryu, Gunma 376-8515, Japan

**Keywords:** EEG signal, MUSIC, PSD, feature extraction, classification, emotion recognition

## Abstract

Electroencephalogram (EEG)-based emotion recognition is a computationally challenging issue in the field of medical data science that has interesting applications in cognitive state disclosure. Generally, EEG signals are classified from frequency-based features that are often extracted using non-parametric models such as Welch’s power spectral density (PSD). These non-parametric methods are not computationally sound due to having complexity and extended run time. The main purpose of this work is to apply the multiple signal classification (MUSIC) model, a parametric-based frequency-spectrum-estimation technique to extract features from multichannel EEG signals for emotional state classification from the SEED dataset. The main challenge of using MUSIC in EEG feature extraction is to tune its parameters for getting the discriminative features from different classes, which is a significant contribution of this work. Another contribution is to show some flaws of this dataset for the first time that contributed to achieving high classification accuracy in previous research works. This work used MUSIC features to classify three emotional states and achieve 97% accuracy on average using an artificial neural network. The proposed MUSIC model optimizes a 95–96% run time compared with the conventional classical non-parametric technique (Welch’s PSD) for feature extraction.

## 1. Introduction

Feelings and emotions have been the sole influence in initiating progress [1]. These important aspects of human behavior, particularly emotion, are highly correlated with human consciousness. Each of them is indispensable for the co-existence of the other [2]. The phenomenal evolution of medical imaging modalities in recent times has facilitated us to explain more of such complex investigations [3]. As a consequence, relevant fields that focus on the application of the theoretical and hypothetical aspects of neuroscience have experienced immense advancements. In the process, the fields of HCI (human-computer interaction), BCI (brain-computer interaction), and neuromarketing [4] have experienced massive breakthroughs. In HCI research, EEG emotional and attention analysis has been a decisive framework for assessing the competence of m-learning platforms [5]. By evaluating learners’ mental states, such a system tweaks the learning environment for optimum efficiency [6]. Modern BCI technologies are capable of ensuring safe driving, as a driver’s EEG features provide highly correlated information regarding their stress levels and mental states [7]. EEG emotional recognition techniques are also being used to evaluate clinical therapies for cancer patients [8]. Wearable EEG headgear is currently revolutionizing the mobile gaming industry by providing a medium for user interaction with virtual reality-based systems. Such interaction is solely based on the analysis of users’ emotional states and motor-imagery states. 

Research in consumer neuroscience has demonstrated that EEG recordings of consumers’ emotional states can provide highly correlated information regarding their product preferences and liking. Such knowledge describing consumer behavior is now being portrayed as a decisive aspect of contemporary marketing schemes [9]. In EEG classification tasks the major challenge lies within the dilemma of feature extraction. Raw time-series signals provide little correlated feature information, yet they hold colossal varieties of hidden feature patterns. Frequency domain transformations, in particular, discrete wavelet transformations (DWT), short-time Fourier transforms (STFT), non-parametric classical power spectral estimators, and parametric-based autoregressive power spectral estimator models can be considered as the state of the art in EEG feature-extraction tasks. 

Authors of [10,11], calculated differential entropy for *δ*, *θ*, *α*, *β*, and *γ* frequency bands for selected channels, following 512 points STFT with non-overlapping windows. Extracted features had the best prediction results in the hybrid model incorporating the Deep Belief Network (DBN) and Hidden Markov Model (HMM). G. Zhao et al. [12] explored both STFT and PSD estimations for extracting features from seven classes of emotional states. For the feature-extraction task, Q. Gao et al. [13] experimented with a unique method of feature fusion. It fuses features from PSD estimations and features extracted from DWT. E. S. Pane et al. [14] used hybrid feature-extraction techniques, which use a non-parametric Welch model for PSD estimation. Z. Yin et al. [15] extracted subject-independent features from PSD estimations and statistical parameters. M. A. Rahman et al. [16] first extracted channels from the frontal, prefrontal, central, parietal, and occipital lobe, by excluding the rest of the channels from each trial of 64 channels. Then they reduced the selected 35 channels to 5 channel data using principle component analysis (PCA). The method used the classic Welch PSD-estimation model for transformation and extracted statistical features from the transformed spectra. 

As the literature review so far suggests, non-parametric classical estimator models are widely used for PSD estimation-based feature extraction from EEG signals. EEG signals can be contemplated as a random process with stationary intervals. Theoretically, Fourier transform (FT) for such signals does not exist, as random processes possess finite power rather than finite energy. However, FT of the autocorrelation function derives the PSD of such random processes. Non-parametric Bartlett, Welch, and Blackman–Turkey estimators are just extensions of this core concept, by introducing non-overlapping or overlapping windowing and averaging techniques. On the contrary, PSD estimators which use parametric Eigen analysis models solely use linear transformations rather than computing windowed periodograms from auto correlated functions [17].

Non-parametric PSD-estimation models come with an expensive computational cost, as they have higher computational complexity and considerably extensive run time. Such extensive run time minimizes the dynamics of HCI and BCI applications, by shrinking real-time data processing capabilities in systems with comparatively fewer configurations [18,19,20]. On the contrary, parametric Eigen analysis-based PSD-estimation models use solely linear transformation and as a result, demonstrate lower computational complexity. On this basis, parametric Eigen decomposition-based techniques such as, the Multiple Signal Classification (MUSIC) model holds potential for efficient spectrum feature extraction from random time series signals. Nevertheless, such parametric-based PSD-estimation models are relatively unexplored in EEG feature-extraction tasks. Specifically, for the widely used SEED [11,21] EEG waveform dataset for emotion recognition no prior researches are available, that investigates the performance of parametric Eigen analysis-based PSD-estimation models in feature extraction.

One reason behind the unpopularity of the parametric PSD estimation is, such models, require some preceding information regarding the signal. For instance, the MUSIC algorithm demands estimations about the signal’s subspace dimension and noise thresholding parameters. In terms of complex random signals analogous to EEG, such parametric information is ambiguous and generates complexity.

The MUSIC algorithm has given incredible results in detecting the direction of arrival (DoA) through PSD estimation, specifically in radar and communication research [22,23]. Previously, the MUSIC model has been very successful in EEG-source localization, due to its efficiency in DoA estimation [24,25]. However, as a consequence of its complexity in terms of PSD estimation of random signals, a limited amount of research has been conducted to investigate its possible use in EEG feature-extraction tasks [26,27]. Particularly in terms of emotional feature extraction, almost no research can be seen which focuses on the possible implementation of the MUSIC model. 

In this paper, we investigated the use of the MUSIC model in EEG emotional feature-extraction tasks, on a publicly available SEED dataset. The feature-extraction model uses parametric-based Eigen-matrix decomposition for estimating power spectral density. Our research also explains the complex concept of estimating subspace dimensions for EEG waveforms in unique cases through detailed analysis. As estimation of subspace dimensionality is indispensable in implementing any Eigen analysis-based parametric PSD estimations, we believe our findings will benefit a wide spectrum of future EEG information-processing research. An enormous amount of research has already been conducted on the SEED dataset; unfortunately, no research has focused on a thorough investigation of the attribute and characteristics of the dataset. Our investigation has also revealed some fascinating attributes of the SEED dataset, which had unknowingly affected previous research works. The novelties of this research work are specifically stated as follows:
Finding technical flaws in the SEED dataset that have not been previously discussed by any research work;Implementing an Eigen decomposition-based parametric feature-extraction model in EEG signal;Proposal of utilizing the MUSIC model for PSD calculation from EEG signals;Run-time comparison between the proposed and conventional PSD estimation;Comparison of the emotional state classifications with other works in the same dataset.

The paper is organized as follows. Section 2 illustrates a detailed overview of the dataset. Section 3 describes the detailed methodology including signal preprocessing techniques, MUSIC model-based feature extraction, and classification. Section 4 presents the results with discussions. Finally, Section 5 presents the concluding remarks for this work.

## 2. Dataset

### 2.1. Dataset Description

This work used a publicly available EEG dataset, which is widely known as the SEED dataset. The SEED dataset is prepared by the BCMI laboratory of Shanghai Jiao Tong University. This EEG dataset is of three emotional states of 15 participants and these states are neutral, positive, and negative emotional states. The distinct emotional states were evoked by showing film clips of each subject. Each film clip is easy to interpret and evokes a single desired targeted emotion. All subjects underwent 15 trials for each of the emotional states. The total elapsed time for EEG waveform acquisition in each trial was 305 s, with each trial representing 62 channel EEG waveforms. 

For each trial, data were down-sampled to 200 Hz and then filtered by a low-pass (cut-off 75 Hz) filter. After down-sampling and filtering, each trial retained the information of 0 Hz to 75 Hz band EEG waveform of 62 channels. The SEED dataset has a total of 675 trials, with 225 trials representing each of the three classes of emotional states. The detailed experimental setup for preparing the dataset is available at [11], and [21].

### 2.2. Dataset Examination

Before going to further processing, our team thought to go through the dataset to examine its channel-wise distribution of signal-amplitude level. Three-dimensional contour plots of the raw time-series data revealed intriguing characteristics of the dataset. It was found that 17% of the total 675 trials contained at least one overshooting channel out of a total of 62 channels. EEG-evoked potential ranged from 10 µV to 100 µV; on the other hand, EMG-evoked potential was at the millivolt level [28]. However, some of the channels exhibited evoked potentials that ranged up to 5 mV. The existence of such overshooting channels could have resulted from electrode leakage or any other arbitrary issue during data acquisition. 

Three-dimensional contour plots of faultless trials from subject 4 are shown in Figure 1 and Figure 2, which represent three-dimensional contour plots of trials with overshooting/corrupted channels. In Figure 1, evoked potentials range up to 200 µV for all the emotional states, which are within the evoked neural potential range. At the same time, the contour plots for the time-series signals are distinguishable for the three psychological states. In Figure 2a, channel 45 (P1) contains recorded evoked potentials of 2500 µV or 2.5 mV, and channel 56 (PO6) contains recorded evoked potentials that range up to 10,000 µV or 10 mV, which lies within the voltage band of EMG signals. The maximum value of overshooting channels’ voltage bands ranges from 0.6 mV to 10 mV, where generally an EEG signal has the amplitude in the microvolt range. Trials from subject 1 demonstrated the faultiest trials with 100% corrupt channels in the negative state and the neutral and the positive states hold 60% and 66%, respectively. Subject 9 demonstrated the least faulty trials with no trials containing any corrupt channels. Table 1 represents the total distribution of trials with overshooting corrupt channels. 

## 3. Material and Methods

The work process of the conducted research is briefly explained in Figure 3. Trial separations for each of the subjects are conducted manually from the parent database. Before pre-processing, detailed analysis of the filtering techniques is conducted to derive the optimal method. Signal filtering is a computationally expensive operation, and such analysis is indispensable in reducing the run time for feature-extraction tasks. The MUSIC model requires a specific estimation for the number of signal subspaces. Such a requirement makes the model difficult to implement in terms of non-stationary signals with random characteristics. The algorithm analysis explores such complexities in terms of implementing the MUSIC model for feature-extraction tasks. The pre-processing includes the implementation of derived optimal signal filters and time-series-dimensionality reduction. Multiple methods are introduced in each of the steps. The methods are tested on bilayer ANN networks, which are built on Keras sequential and functional APIs. Each of the steps is discussed briefly here.

### 3.1. Signal Filtering

Before the extraction of a specific psychological information band, a 2nd order Butterworth IIR bandstop filter with a half-power frequency band of 48 Hz to 52 Hz is implemented as a notch filter to remove power-line noise. Eyeblink artifacts are removed with the help of the EAWICA toolbox. This toolbox-utilization procedure has been widely explained [29,30]. The raw signal is then filtered to extract the relevant neural bands holding emotional state information. To deduct an optimal and time-efficient filtering method FIR Hamming window, FIR Kaiser window, and FIR Chebyshev window are tested. For the 4 Hz to 40 Hz band, the Chebyshev window has a sharper cutoff. On the other hand, the Kaiser window and Hamming window demonstrated similar performance. However, as the Hamming window possesses a better time-complexity function, a higher-order FIR Hamming window of 500 data points was chosen. EEG waveform holds 5 major bands, which are *δ*, *θ*, *α*, *β*, and *γ* wave. As the *δ* band represents sleep waves and the *θ* band represents the deeply relaxed phase, both of them can be ignored for information extraction in this particular case. Our bands of interest should only represent attention and concentration states, as they are relevant to the experiment. As a result, only the *α*, *β*, and *γ* bands are our targeted extraction bands. 

### 3.2. MUSIC Model

The concept of signal and noise subspace is crucial for understanding the fundamentals of the MUSIC algorithm. The vector $\left[x_{1}(n)\;x_{2}(n)\;x_{3}(n)\;\ldots \;x_{p}(n)\right]$ can be considered as the signal-subspace vector of an arbitrary signal *y*(*n*) if,
(1)$$y(n)\equiv [x_{1}(n)\;x_{2}(n)\;x_{3}(n)\;\ldots \;x_{p}(n)].$$

If $\left[x_{p+1}(n)\;x_{p+2}(n)\;x_{p+3}(n)\;\ldots \;x_{L}(n)\right]$, the vector represents the noise subspace for the signal, then the signal *y*(*n*) can be presented as,
(2)$$\begin{array}{ll} y(n)= & x_{1}(n)+x_{2}(n)+\ldots +x_{p}(n)+\\ & x_{p+1}(n)+x_{p+2}(n)+\ldots +x_{L}(n) \end{array}$$

Here p represents the number of signal subspaces. The MUSIC algorithm estimates the PSD of the targeted signal *y*(*n*) by decomposing the signal autocorrelation function, $\Gamma_{yy}$, to its auto-correlated function of signal subspace, $\Gamma_{xx}$, and noise subspace $\sigma_{w}$. The signal subspace and the noise subspace are orthogonal to each other. The concept can be mathematically expressed as,
(3)$$\Gamma_{yy}=\Gamma_{xx}+\sigma_{w}^{2}I.$$

The dimension of $\Gamma_{yy}$ is L×L and rank is p (number of subspace), where p<L. (3) can be further simplified into,
(4)$$\Gamma_{yy}=\sum_{i=1}^{p}\lambda_{i}\nu_{i}\nu_{i}^{H}+\sigma_{\omega }^{2}\nu_{i}\sum_{i=1+p}^{M}\nu_{i}\nu_{i}^{H}.$$

Here $\lambda_{i}$, represents the non-zero eigenvalue for the range of i=1 to p. Moreover, $v_{i}$ is the eigenvector and $v_{i}^{H}$ is the Hermitian conjugate of the eigenvector. The term $\lambda_{i}v_{i}v_{i}^{H}$ represents each signal subspace element of *y*(*n*). Equation (4) fully decomposes the targeted signal and is considered a fundamental of the MUSIC model. The targeted signal can be represented as the complex sinusoidal vector s(ω),
(5)$$s(\omega )\equiv [1\;e^{j\omega }\;e^{2j\omega }\;\ldots \;e^{\left(M-1\right)j\omega }].$$

The eigenvectors in noise subspace $v_{noise}$ are denoted as,
(6)$$v_{noise}\equiv [v_{p+1}\;v_{p+2}\;v_{p+3}\;\ldots v_{k}].$$

Then the MUSIC model frequency estimator $P_{MUSIC}(\omega )$ is stated as,
(7)$$P_{MUSIC}(\omega )=\frac{1}{\sum_{k=p+1}^{M}\left|s^{H}(\omega )v_{k}\right|}.$$

Here, $s^{H}(\omega )$ is the Hermitian conjugate of the vector s(ω). After calculating the peaks (estimated frequencies) of $P_{MUSIC}(\omega )$, the power spectrum can be calculated from the following Pisarenko harmonic equation [31],
(8)$$\left[\begin{array}{cccc} \cos\omega_{1} & \cos\omega_{2} & \cdots & \cos\omega_{p}\\ \cos2\omega_{1} & \cos2\omega_{2} & \cdots & \cos2\omega_{p}\\ . & . & \cdots & .\\ . & . & \cdots & .\\ \cos p\omega_{1} & \cos p\omega_{2} & \cdots & \cos p\omega_{p} \end{array}\right]\left[\begin{array}{c} P_{1}\\ P_{2}\\ .\\ .\\ P_{p} \end{array}\right]=\left[\begin{array}{c} \gamma_{yy}(1)\\ \gamma_{yy}(2)\\ .\\ .\\ \gamma_{yy}(p) \end{array}\right].$$

Vergallo et al. [32] implemented the MUSIC algorithm for brain-source localization and their findings suggest that the neural signal band consists of three source dipoles. As per their research, the number of signal subspaces depends mostly on the extracted information bands for an arbitrary EEG channel. The vector $\left[\begin{array}{cc} \begin{array}{cc} \begin{array}{cc} \begin{array}{cc} \delta & \theta \end{array} & \alpha \end{array} & \beta \end{array} & \gamma \end{array}\right]$ can be generalized as the subspace vector for an arbitrary channel holding all neural information bands. However, as discussed previously, in this particular case our targeted neural information bands only lie in the range of 8 Hz to 40 Hz. For this, the number of subspaces, p in this particular case, can be estimated as 3 and the subspace vector can be denoted as $\left[\begin{array}{cc} \begin{array}{cc} \alpha & \beta \end{array} & \gamma \end{array}\right]$.

The estimated subspace vector is also clarified in Figure 4. In Figure 4a the three peaks clearly state the presence of *α*, *β*, and *γ* bands, as the peaks can be found near 10 Hz, 15 Hz, and 20 Hz for all the states. All five bands are considered for calculating the spectrum in Figure 4b, but the spectrum does not demonstrate any peaks near *δ* and *θ* bands (1 Hz to 8 Hz). This clarifies the absence of these two information bands in the target signal. The number of the subspace is undefined (p=∞) in Figure 4c and the spectrum is calculated by only thresholding the eigenvalues for the noise subspace. The spectrum for the negative state gives three major peaks which lie in *α* (peak at 10 Hz), *β* (peak at 27 Hz), and *γ* (peak at 35 Hz) bands. Each of the state spectrums in Figure 4a is distinguishable and clarifies the estimated signal-subspace vector for the targeted EEG channels. Thus, for the whole dataset transformation, the MUSIC spectrum is calculated considering p=3. 

### 3.3. Pre-Processing and Feature Extraction

Figure 5 and Figure 6 explain the pre-processing and feature-extraction method which excludes the corrupt/overshooting channels. Following the literature, in [16] only prefrontal, frontal, central, parietal and occipital channels are extracted, excluding the rest of the 22 channels. Each channel band matrix (M×N, M represents the number of channels and N represents the number of data points) is then filtered for handling baseline noise and eye-blink artifact removal.

After the initial filtering corrupt/overshooting channels are excluded for each M×N channel-band matrix, followed by an 8 Hz to 40 Hz bandpass FIR filter for extracting the relevant information band. Each clean channel-band matrix is then reduced to a single channel (1×N dimensional vector) using PCA. Each PCA vector is then cascaded to form the reduced 5×N dimensional trial vector. The workflow is designed to gain maximum computational efficiency. Lastly, Figure 7 demonstrates the pre-processing and feature-extraction methods, which include the overshooting channels. The pre-processed matrices are then transformed using a MUSIC model with 50% overlapping data points.

### 3.4. Training and Method Evaluation

For training and evaluation, the k-fold cross-validation technique is followed, where the value of k is considered as 5. The dataset is split into 5 equal folds, and then the networks are repeatedly trained on 4 folds and tested on the remaining fold. The average accuracy from the five training: and testing incidents are evaluated; due to folding each trial gets the chance to appear in both the training and testing set at least once. Moreover, in each of the folds, 10% of the training data is used as the validation set.

Moreover, in each of the folds, 10% of training data is used as the validation set. Each feature set is trained on a fully connected ANN, which has a 3-node SoftMax layer acting as the final classification layer. The network uses the ReLU activation function, and binary cross entropy (BCE) loss is used as the loss function for optimization. The ANN used in the experiment has two hidden layers, consisting of 512 and 248 nodes, respectively. In each of the experiments the network is trained for a maximum of 500 epochs, but if the validation loss is not improved for 20 consecutive epochs, the training is terminated. The initial learning rate for the experiments is 0.001, with an epoch patience of 5. If for 10 consecutive epochs validation loss is not improved, then the learning rate is increased by 25%. 

## 4. Results and Discussion

For the feature set constructed from the 62×N dimensional pre-processed signal matrix (considering all 62 channels), subject wise 5-fold cross-validation accuracy from the network is 97%. As our additional findings suggest, specific trials from the seed dataset hold corrupted channels demonstrating voltage bands, which are much higher than desired EEG-voltage bands. These channels can be considered EMG artifacts or caused due to electrode-leakage current, and do not demonstrate any EEG information. Until now, no research has handled these specific channels demonstrating corrupt data. As discussed earlier for such particular cases, we have excluded the corrupt/overshooting channels and then reduced the number of channels with PCA. In this case, the feature space is constructed from a 5×N dimensional reduced signal matrix. From this distinct feature space, which excludes the corrupted channels, subject-wise 5-fold cross-validation accuracy from the network is 86.53 %.

Our research also implements higher dimensional visualization techniques for interpreting the influence of the corrupted channels in the feature space. Figure 8 represents a three-dimensional visualization of the constructed MUSIC feature space for a trial demonstrating the negative stage. In Figure 8a, the feature space is constructed from the 62×N dimensional signal matrix, which includes the corrupted channels for this particular trial. In Figure 8b,c, the feature space is constructed from the 5×N dimensional reduced signal matrix, but only the latter representation excludes the corrupted channels through peak finding. The overshooting channels for this particular trial lie in the parietal and occipital bands. Correspondingly, in the 62×124 dimensional MUSIC feature space for the trial, spectrum values for the rest of the channels are suppressed due to much higher peaks generated from the overshooting channels in the *β* band. In the feature space extracted from the reduced signal matrix, which also includes these overshooting channels, the *PCA_occipital_* and *PCA_parietal_* bands similarly suppress the spectrum from the rest of the bands. Removing the corrupt/overshooting channels from the occipital and parietal bands provide the true spectrum distribution across all PCA bands, which are represented in Figure 8c. 

Subsequently, it can be observed in Figure 8a,b, that the presence of the corrupt channels generates high peaks which demonstrate β band components. When the particular overshooting channels are excluded, the β components are significantly reduced and the α band is dominating across all channels. Additionally, in such a case the presence of the *γ* band can also be observed, which is absent when the corrupted channels are present in the feature space. The high amplitude peaks in the *β* bands suggest the presence of EMG artifacts in the trial due to the corrupted channels. The corrupt/overshooting channels demonstrate similar effects across all subjects. The spectrums of Figure 9 and Figure 10 represent feature space for subject 12 from the MUSIC spectrum and non-parametric (Welch) spectrum, respectively. For Figure 9a and Figure 10b, the weight of the α band is significantly higher across all the spectrums. 

The MUSIC spectrum visualizations interpret that, α band is dominant for each of the states but varies with amplitude. For the frontal and central channels, *γ* components are also observed for the neutral and positive states. Both 62×124 and 5×124 dimensional feature space demonstrates the presence of similar signal subspace bands. 

The first three principal components hold around 65% to 95% of the information for the entire feature space; consequently, three-dimensional visualizations of these three principal components can provide explicit information. The PCA visualizations in Figure 11 and Figure 12 demonstrate the quality of each of the feature spaces, as the visualizations can interpret class separability. In this regard the MUSIC model also demonstrates superiority over the classical non-parametric approach of feature extraction, for both of the signal pre-processing approaches. Figure 11a represents the MUSIC feature space and Figure 11b represents the non-parametric (Welch) feature space, from the pre-processed 62×N dimensional signal matrix. Figure 12a,b represents MUSIC and Welch feature spaces, respectively, for the 5×N dimensional reduced signal matrix. Both of the figures’ feature spaces that are generated from the MUSIC model form more separable clusters. As demonstrated in the visualization, Welch feature space of subject 1, clearly cannot distinguish neutral and positive states when all 62 channels are considered. 

From comparing Figure 11a and Figure 12a, it is evident that the quality of the feature space is significantly decreased when the number of channels is reduced. The subject-wise accuracies from k-fold cross-validation also clarify such behavior. The subject-wise 5-fold cross-validation accuracies from the MUSIC-generated feature spaces are shown in Figure 13. For the feature set that is constructed without channel exclusion, the network demonstrates 100% accuracy for 6 out of 15 subjects and achieved over 97% classification accuracy for 13 out of 15 subjects. The average subject-wise accuracy is 97% for this specific technique. For the feature set that excludes the overshooting/corrupt channels before signal pre-processing, the network demonstrates a subject-wise average accuracy of 86.53%. In this particular case, the networks perform satisfactorily across all subjects except subjects 4 and 9. When subject 4 and subject 9 are excluded, the average accuracy is observed to be over 90% for this particular channel-exclusion technique. 

Subject-wise precision, recall, and F1 scores for both of the networks on the feature set that do not exclude any channels are illustrated in Figure 14. The exceptionally high-precision values suggest that the network is extremely efficient in predicting true values for each of the states across all subjects. All of the average metric values are over 96.5%, which suggests that both the feature space and the networks demonstrate excellent performance in each of the state-classification tasks.

Subject-wise precision, recall, and F1 score for the feature space that excludes the overshooting/corrupt channels before reducing the number of channels with PCA, are shown in Figure 15. The matrices suggest that for this specific technique the network performs comparatively better in predicting the negative states.

The k-fold validation results and the feature-space visualization techniques both suggest that by reducing the number of channels the quality of the MUSIC feature space is significantly reduced. The network performance on the 45×62×124 dimensional feature space is exceptionally higher than the feature space constructed by reducing the number of channels. 

Compared with existing high-performance classification techniques on this particular SEED dataset, our 97% subject-dependent classification accuracy suggests that the proposed MUSIC model is superior for such a feature-extraction task. Table 2 makes a detailed comparison with existing works. M. A. Rahman et al. [16] explore Welch-based brain topography for feature extraction and CNN for the classification task, which achieved around 94% accuracy. A. Bhattacharyya et al. [33] explored modified wavelet-based decomposition for feature extraction and auto-encoder-based random forest for classification, this particular approach also achieved an accuracy level of around 94%. W. Zheng et al. [34] used parametric PSD-estimation models and F. Wang et al. [35] used STFT for the emotional feature-extraction task, which achieved 86.65% and 90.59% accuracy, respectively.

As is evident from the comparison, our proposed approach achieves significant improvement. Additionally, none of the existing research has made any investigation considering corrupt channels. As discussed previously, the dataset contains a large number of trials, which holds multiple corrupt/overshooting channels. These overshooting channels were ignored in previous research on this dataset. Nevertheless, due to the existence of these particular channels in the feature space, the network will demonstrate bias. Such bias will affect network performance on an external dataset, which does not hold such flawed channels. Thereby these specific channels should be excluded before feature extraction. Our investigated approach of removing these specific channels before constructing the feature space has demonstrated 86.53% accuracy on the network. This article is the first research to observe such extensive flaws in the SEED dataset. The results from the computation-time analysis are shown in Figure 16.

The figure illustrates the computation time for the MUSIC model and non-parametric model-based feature-extraction task, along with the execution time for both of the signal pre-processing methods investigated in this research. The computation time varies intensively with the length of the targeted matrix and across machine specifications. Each of the tasks was executed in an intel CORE-i5 processor with 32 GB DDR4 RAM, 512 GB SSD Memory, and 4 GB NVIDIA GEFORCE GTX 960 M graphics card, under a prioritized thread. The length of each trial was duly considered for the computation-time analysis. Table 3 summarizes the run-time analysis, which shows non-parametric feature-extraction techniques. Although the pre-processing techniques consider all 62 channels to enhance the quality of feature space, they are computationally heavy.

According to the results from computational time analysis, the MUSIC spectrum exhibits far greater superiority in terms of run time. Even for a full 62-channel trial, the computation time was investigated to be near 1.3 s. On the contrary, the non-parametric estimation models required around 28 s to 40 s in a prioritized thread. As Table 3 illustrates, the MUSIC model can optimize computation time for feature-extraction tasks by around 94% to 96%. With a PCA-based pre-processing approach, the feature-extraction time is just around 0.1 s. The results also suggest that with the implementation of the MUSIC model along with PCA-based pre-processing techniques, an EEG-based BCI system can achieve comparatively real-time performance.

## 5. Conclusions

Owing to the complexity of implementing the MUSIC algorithm for random signals, the algorithm is rarely investigated in feature-extraction tasks for neural signals. In this research, we investigated the performance of the MUSIC algorithm in neural information processing by implementing the MUSIC model-based feature-extraction method for emotional state-recognition tasks from multichannel EEG recordings. With a bi-layer ANN network, the MUSIC model-generated feature space achieves an admirable 97% accuracy on the SEED emotional dataset. Our investigation also finds that a significant number of trials in the SEED dataset hold corrupt or overshooting channels which were previously overlooked, and we have also investigated the effects of these channels on subject-dependent classification tasks. Future research on this dataset should benefit from such intriguing observation. The existence of the corrupt channels in the SEED dataset could have resulted from electrode leakage or any other arbitrary issue during data acquisition. State-of-the-art filtering techniques such as dynamic spatial filtering or adaptive filtering, which uses weighted attention to extract features from good channels leaving out the corrupted channels, can be investigated on this dataset for future work. We achieved a subject-dependent emotional state classification accuracy of 86.53% after excluding these specific channels. We then compared the quality of the MUSIC model-generated feature space with feature space generated by implementing conventional methods. Finally, our computation-time analysis of various methods illustrates that the MUSIC model can optimize 94% to 95% run time in similar feature-extraction tasks. Such findings suggest that the MUSIC algorithm has great prospects in real-time BCI applications.

## Figures and Tables

**Figure 1 bioengineering-10-00099-f001:**
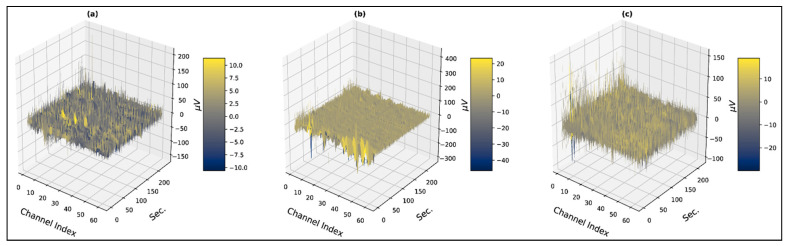
Three-dimensional visualization of faultless trials (**a**) subject: 4, state: negative, trial: 3 (**b**) subject: 4, state: neutral, trial: 15 (**c**) subject: 4, state: positive, trial: 13.

**Figure 2 bioengineering-10-00099-f002:**
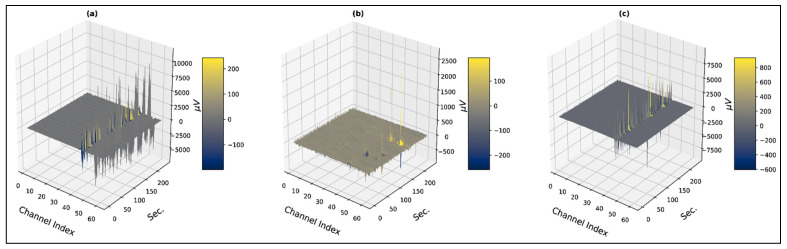
Three-dimensional visualization of faulty trials with overshooting/corrupted channels (**a**) subject: 1, state: negative, trial: 3 (**b**) subject: 1, state: neutral, trial: 4 (**c**) subject: 1, state: positive, trial: 15.

**Figure 3 bioengineering-10-00099-f003:**
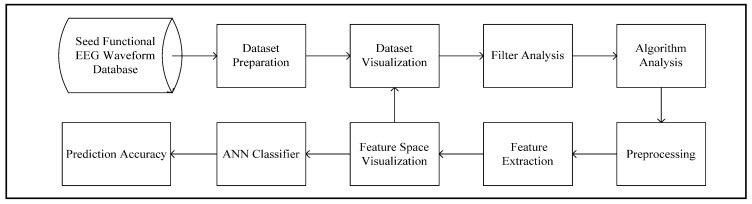
Flow diagram explaining research methodology in brief.

**Figure 4 bioengineering-10-00099-f004:**
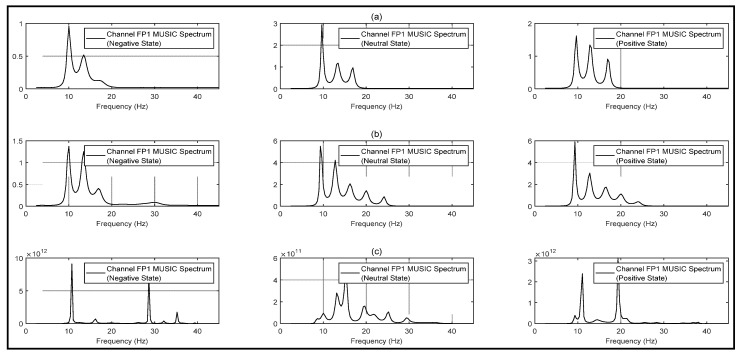
MUSIC-transformed power-spectrum density spectrum with varying signal subspace estimation for subject: 15, state: negative, trial: 10, channel: FP1 (**a**) considering *δ*, *θ*, *α*, *β* and *γ* bands as signal subspace vector (**b**) considering α, β and γ bands as signal subspace vector (**c**) undefined signal subspace vector.

**Figure 5 bioengineering-10-00099-f005:**
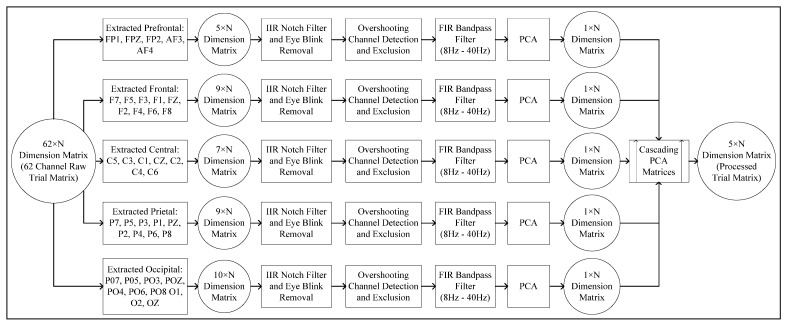
Flow diagram for channel exclusion and dimensional reduction.

**Figure 6 bioengineering-10-00099-f006:**
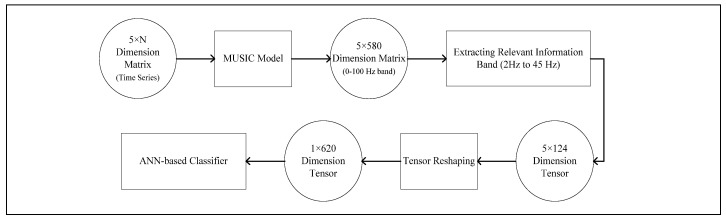
Flow diagram explaining feature-extraction process for 5 *×* 124 Tensor from reduced 5 *× N* dimension trials.

**Figure 7 bioengineering-10-00099-f007:**
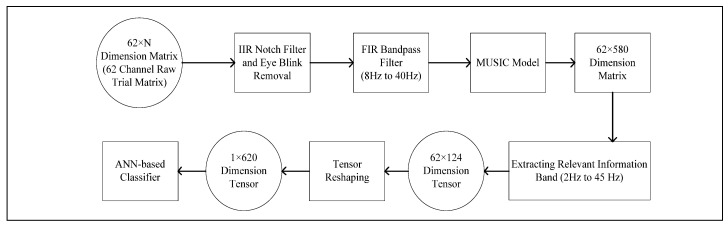
Flow diagram explaining feature-extraction process for 62 *×* 124 Tensor from raw 62 *× N* dimension trials.

**Figure 8 bioengineering-10-00099-f008:**
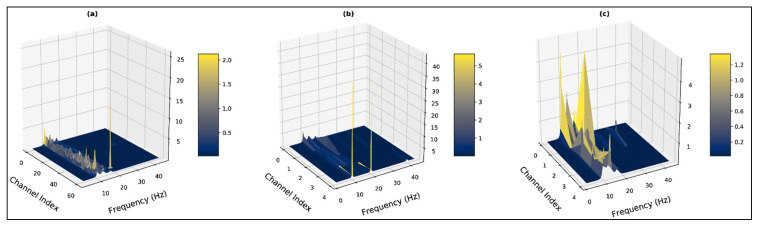
Three-dimensional visualization of the computed MUSIC spectrums for subject: 1, state: negative, trial: 5 (**a**) considering all 62 channels (**b**)reducing the number of channels (**c**) excluding the corrupt channels before channel reduction.

**Figure 9 bioengineering-10-00099-f009:**
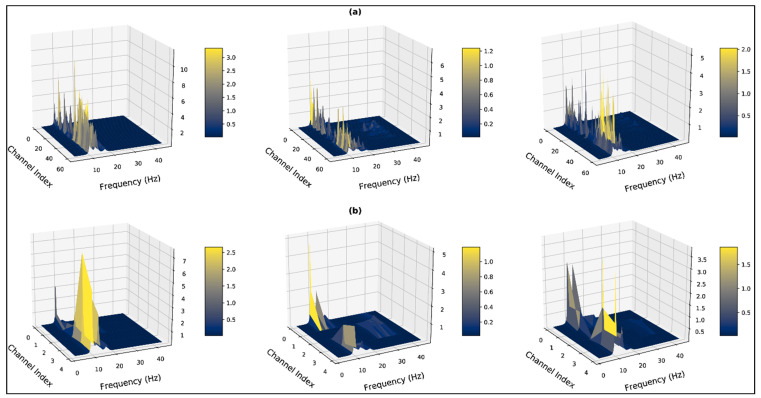
Three-dimensional visualization of the computed MUSIC spectrum for the negative state (left), the neutral state (middle), and the positive state (right) of subject:12, trial: 14 (**a**) considering all 62 channels (**b**) excluding the corrupt channels before channel reduction.

**Figure 10 bioengineering-10-00099-f010:**
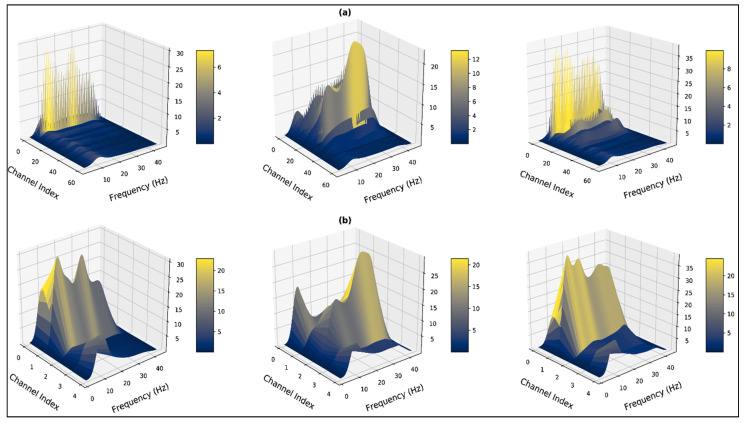
Three-dimensional visualization of non-parametric (Welch) spectrum for the negative state (left), the neutral state (middle), and the positive state (right) of subject:12, trial: 14 (**a**) considering all 62 channels (**b**) excluding the corrupt channels before channel reduction.

**Figure 11 bioengineering-10-00099-f011:**
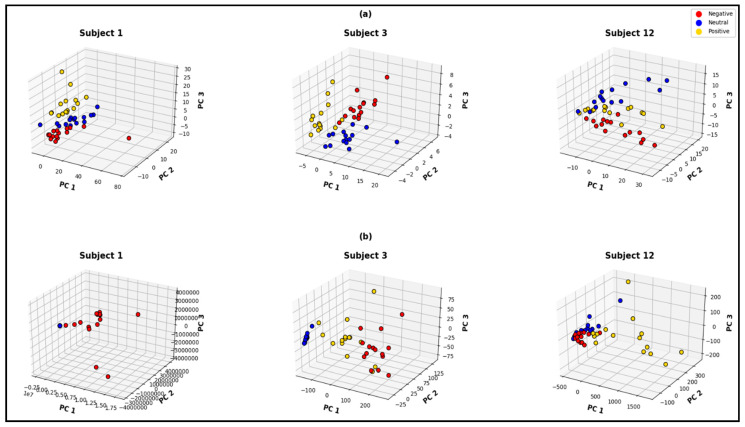
Three-dimensional principal-component visualization of (**a**) the computed MUSIC feature space and (**b**) non-parametric (Welch) feature space considering all 62 channels.

**Figure 12 bioengineering-10-00099-f012:**
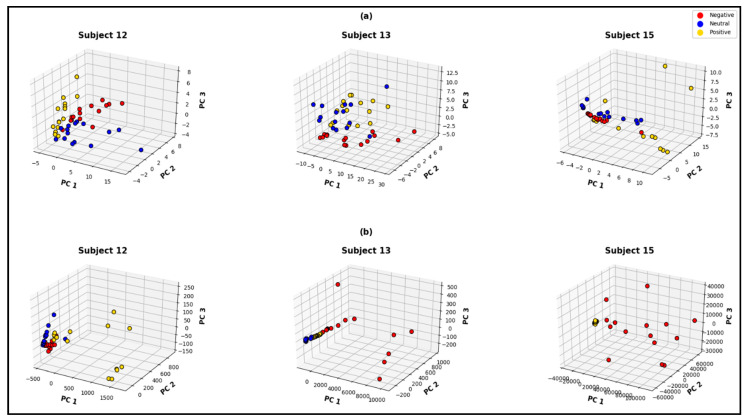
Three-dimensional principal-component visualization of (**a**) the computed MUSIC feature space and (**b**) non-parametric (Welch) feature space after excluding the corrupt channels before channel reduction.

**Figure 13 bioengineering-10-00099-f013:**
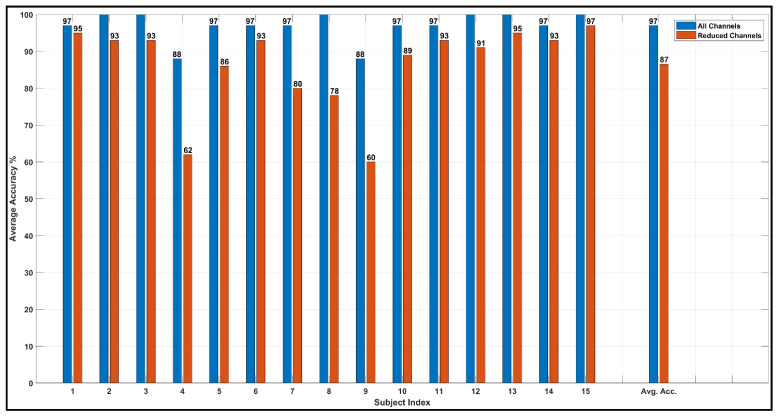
Subject wise 5-fold cross-validation accuracy from MUSIC feature space.

**Figure 14 bioengineering-10-00099-f014:**
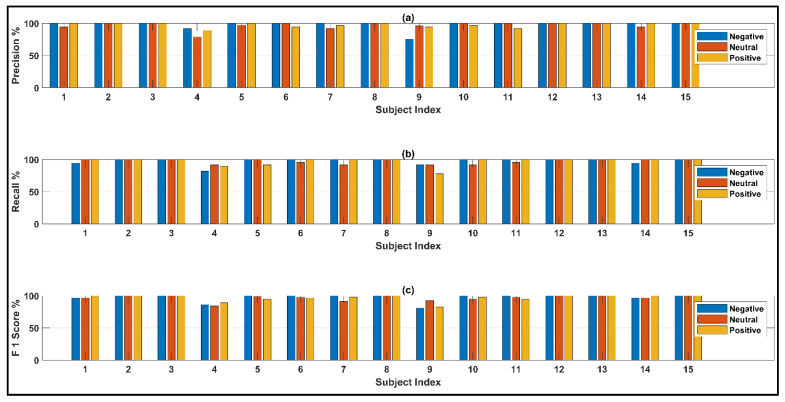
Observed (**a**) precision (**b**) recall and (**c**) F1 scores from the network considering all 62 channels.

**Figure 15 bioengineering-10-00099-f015:**
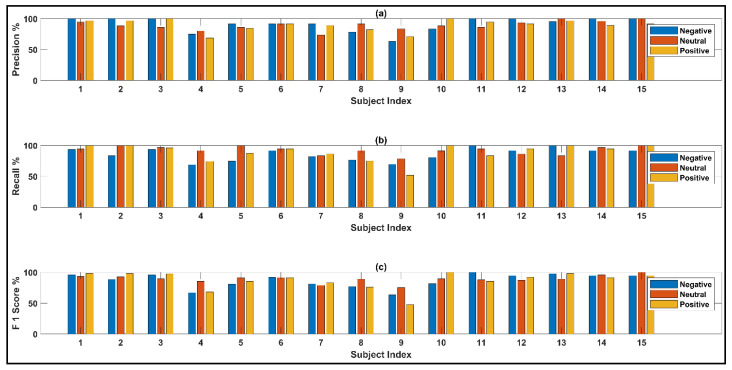
Observed (**a**) precision (**b**) recall and (**c**) F1 scores from the network excluding the corrupt channels.

**Figure 16 bioengineering-10-00099-f016:**
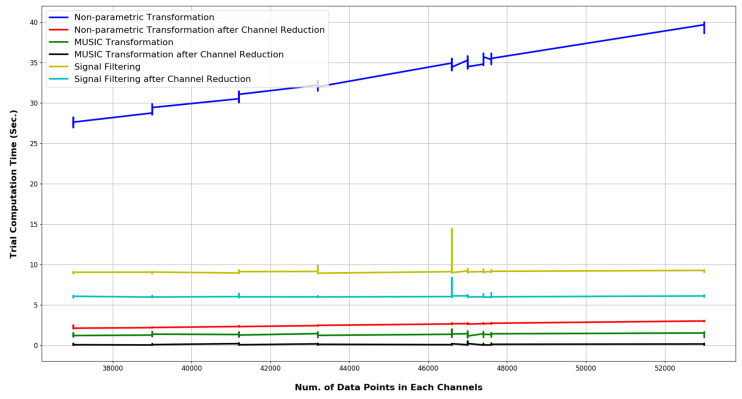
Run time comparison of various methods for multi-channel EEG pre-processing and feature-extraction tasks.

**Table 1 bioengineering-10-00099-t001:** Distribution of trials with overshooting corrupt channels.

State	Sub 1	Sub 2	Sub 3	Sub 4	Sub 5	Sub 6	Sub 7	Sub 8	Sub 9	Sub 10	Sub 11	Sub 12	Sub 13	Sub 14	Sub 15
Negative	15	2	1	1	6	12	0	0	0	1	0	0	8	3	15
Neutral	9	0	0	0	0	2	2	0	0	2	0	3	0	1	2
Positive	10	4	1	0	2	2	0	2	0	3	0	3	0	2	3

**Table 2 bioengineering-10-00099-t002:** Performance comparison with existing research on emotional state classification for the seed dataset.

Author and Research	Feature Extraction Method	Classification Method	Average Accuracy
Y. Jin et al. [36]	Differential Entropy	Domain Adaptive Network	79.19%
D. W. Chen et al. [37]	Differential Entropy	Linear Discriminant Analysis	82.5%
W. L. Zheng et al. [21]	Critical Frequency Band Investigation	Deep Belief Network	86.08%
Y. Yang [38]	Differential Entropy	Hierarchical Network	86.42%
M. A. Rahman et al. [16]	PCA and non-parametric Welch model	ANN	86.57%
W. Zheng et al. [34]	Parametric Model	Group Sparse Canonical Correlation Analysis	86.65%
Y. Luo et al. [39]	Data augmentation approach	Generative adversarial network	87%
X. Wu et al. [40]	Connectivity Network	SVM	87%
F. Yang et al. [41]	High dimensional features	ST-SBSSVM	89%
F. Wang et al. [35]	STFT	CNN	90.59%
W. Zheng et al. [42]	Differential Entropy	Discriminative graph regularized	91%
A. Bhattacharyya et al. [33]	Wavelet-based decomposition	Random Forest (Autoencoder based)	94.4%
M. A. Rahman et al. [43]	Welch topographic map	CNN	94.63%
Proposed Method	MUSIC model (Includes all 62 channels)	Bilayer ANN	97%

**Table 3 bioengineering-10-00099-t003:** Investigated computation-time metadata for non-parametric feature-extraction method (Welch model) vs MUSIC model.

Trial Dimension	Channel Length	Computation Time (s)		Optimization %
		Welch Based Model	MUSIC Based Model	
62 × *N*	Below 40,000	28.30 s	1.32 s	95.33%
	40,000 to 45,000	31.09 s	1.35 s	95.65%
	45,000 to 50,000	35.05 s	1.38 s	96.07%
	50,000 to 55,000	39.36 s	1.39 s	96.47%
5 × *N*	Below 40,000	2.37 s	0.12 s	94.94%
	40,000 to 45,000	2.46 s	0.14 s	94.31%
	45,000 to 50,000	2.64 s	0.13 s	95.07%
	50,000 to 55,000	2.79 s	0.13 s	95.34%

## Data Availability

Supporting data are available at https://drive.google.com/drive/folders/1DsH9qlNBRIhw3me94LThHwqc4esPe5Au (accessed on accessed on 28 November 2022).

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
