# Peer review of "Emotional State Classification from MUSIC-Based Features of Multichannel EEG Signals"

_bioengineering, 2023, doi:10.3390/bioengineering10010099_

Round 1
Reviewer 1 Report
Clearly represent the differences between parametric and non-parametric method.
Give the proper reason for selecting the non-parametric method for your study. There are several non-parametric methods are available. If any specific reason for this?
In your paper you have mentioned that Particularly in terms of emotional feature extraction, almost no research can be seen which focuses on the possible implementation of the MUSIC model.
Is it true?
What all are the technical flaws in the SEED dataset you found?
How can you solve that?
literature survey conducted by the authors were not sufficient.
Need more explanation about the SEED dataset. How they prepared the dataset? What are the parameters they arranged in the dataset?
How many features extracted per trials?
What is training and testing data percentage.
Need more explanation for experimental setup.
Result obtained by this study was ok.
Reviewer 2 Report
This paper described the parametric-based frequency spectrum estimation method (MUSIC) to extract power spectral density features of EEG signals for emotion recognition tasks. About 97% of the three emotional states were accurately distinguished by a fully-connected deep learning neural network. The following comments could be considered for improvement:
1) The term MUSIC should be defined when it is first mentioned in the text.
2) The settings of signal filtering procedure and toolbox should be provided in Section 3.1. For example, how to remove the eye blink artifacts by the EAWICA toolbox. What about the parameters of FIR Hamming window, FIR Kaiser window, and FIR Chebyshev window.
3) In Section 3.4, the details of the fully-connected artificial neural network should be provided, such as the network structure, topology, training algorithm, and pooling method.
4) It is better to describe the abbreviations "Neg", "Neu", and "Pos" in the captions of Figure 14 and Figure 15.
5) The typesetting of the article should be adjusted to avoid a table occupying multiple pages.
Round 2
Reviewer 1 Report
All corrections were carried out by the authors.
Author Response
Not Applicable.
Reviewer 2 Report
The revision manuscript has been improved with the corresponding modifications in response to the comments. The manuscript is acceptable for publication with some typesetting adjustment of floating mathematical symbols.
Author Response
Comments and Suggestions for Authors by reviewer:
The revision manuscript has been improved with the corresponding modifications in response to the comments. The manuscript is acceptable for publication with some typesetting adjustment of floating mathematical symbols.
Author's Response: The authors would like to thank the reviewer for his observation regarding some typesetting adjustment in our submitted revised manuscript. According to the suggestions of reviewer, we have gone through the whole manuscript carefully and adjusted the typesetting of floating mathematical symbols.
The revisions made to the manuscript was marked up by using the “Track Changes” function so that the changes can be viewed by editor and reviewer.